SLFCNet: an ultra-lightweight and efficient strawberry feature classification network

Xu Wenchao 1
Wang Yangxu wangyx6432@gmail.com 2 3
Yang Jiahao 3
1 School of Electrical and Computer Engineering, Nanfang College Guangzhou , Conghua , Guangdong , China
2 Department of Network technology, Guangzhou Institute of Software Engineering , Conghua , Guangdong , China
3 College of Robotics, Guangdong Polytechnic of Science and Technology , Zhuhai , Guangdong , China
Coelho Paulo Jorge
Electronic publication date: 2025 Jan 2
Publication date: 2025
Volume: 11
Electronic Location ID: e2085
Received 2023 Nov 10; Accepted 2024 May 5
Copyright: ©2025 Xu et al.
Copyright year: 2025
Copyright holder: Xu et al.
License: This is an open access article distributed under the terms of the Creative Commons Attribution License, which permits unrestricted use, distribution, reproduction and adaptation in any medium and for any purpose provided that it is properly attributed. For attribution, the original author(s), title, publication source (PeerJ Computer Science) and either DOI or URL of the article must be cited.
License URL: https://creativecommons.org/licenses/by/4.0/

Keywords: Strawberry, Lightweight, Detection and classification, Real-time recognition, Automated management

Funding: The authors received no funding for this work.

==============================
Background

As modern agricultural technology advances, the automated detection, classification, and harvesting of strawberries have become an inevitable trend. Among these tasks, the classification of strawberries stands as a pivotal juncture. Nevertheless, existing object detection methods struggle with substantial computational demands, high resource utilization, and reduced detection efficiency. These challenges make deployment on edge devices difficult and lead to suboptimal user experiences.

Methods

In this study, we have developed a lightweight model capable of real-time detection and classification of strawberry fruit, named the Strawberry Lightweight Feature Classify Network (SLFCNet). This innovative system incorporates a lightweight encoder and a self-designed feature extraction module called the Combined Convolutional Concatenation and Sequential Convolutional (C3SC). While maintaining model compactness, this architecture significantly enhances its feature decoding capabilities. To evaluate the model’s generalization potential, we utilized a high-resolution strawberry dataset collected directly from the fields. By employing image augmentation techniques, we conducted experimental comparisons between manually counted data and the model’s inference-based detection and classification results.

Results

The SLFCNet model achieves an average precision of 98.9% in the mAP@0.5 metric, with a precision rate of 94.7% and a recall rate of 93.2%. Notably, SLFCNet features a streamlined design, resulting in a compact model size of only 3.57 MB. On an economical GTX 1080 Ti GPU, the processing time per image is a mere 4.1 ms. This indicates that the model can smoothly run on edge devices, ensuring real-time performance. Thus, it emerges as a novel solution for the automation and management of strawberry harvesting, providing real-time performance and presenting a new solution for the automatic management of strawberry picking.

Introduction

Strawberries stand as one of the world’s foremost crops of significant economic importance, bearing ever-increasing relevance to human daily production and livelihoods (Kolle et al., 2019). In 2021, the global market valuation for fresh strawberries was estimated at approximately 247.9 billion US dollars, with projections foreseeing an ascent to about 433.3 billion US dollars by 2028 (SkyQuest, 2022). However, strawberries are perishable and difficult to store for long periods, making the timely harvesting of ripe strawberries crucial. Yet, traditional manual harvesting methods are inefficient and costly (Thakur et al., 2020). Therefore, the technology for automated and real-time detection of strawberry ripeness is particularly important (Shao et al., 2020), which can not only improve economic benefits but also reduce labor costs.

Fortunately, there has been a significant development in computer vision technologies, prompting researchers to propose numerous deep learning-based object detection algorithm models. These models have been utilized for various crops, including apples (Wang & He, 2021), tomatoes (Cardellicchio et al., 2023), strawberries (Lawal, 2024; Habaragamuwa et al., 2018; He et al., 2023), potatoes (Dai, Hu & Fan, 2022), tea leaves (Yang et al., 2019) , and lychees (Xie et al., 2022), among others.

Tian et al. (2019) designed a system based on YOLOv3 (Redmon & Farhadi, 2018) that can detect three different stages of apples in an orchard in real-time. Yang et al. (2022) proposed a BCo-YOLOv5 network model for fruit object recognition and detection in orchards, introducing a bidirectional cross-attention mechanism (BCAM) into the network, achieving a mean average precision (mAP) of 97.70%. Especially in the field of strawberry monitoring, current research mainly focuses on optimizing single-stage object detection algorithms based on the YOLO series (Redmon et al., 2016). An et al. (2022), inspired by HorNet, designed the C3HB module by integrating Horblock and C3 structures, and then embedded the normalization-based attention module (NAM) into the neck. Li et al. (2023) introduced an adaptive spatial feature fusion (ASFF) module into YOLOv5, enabling the network to adaptively learn strawberry features, but the inference efficiency is low. Wang et al. (2022) proposed a multi-stage method for detecting strawberry fruits based on YOLOv3, with an average precision (mAP@0.5) of 86.58% and an F1-Score of 81.59%. However, although these models have improved accuracy in laboratory environments, their efficiency in identifying the ripeness of strawberry fruits in actual field environments still needs to be improved.

Nevertheless, researchers have also begun to deploy improved models into actual carriers for use in real environments, which have been widely applied in tasks such as recognizing ripeness levels, predicting yields, remote monitoring, and harvesting strawberries. In a recent study, Chai, Xu & O’Sullivan (2023) attempted to deploy the improved YOLOv7 model on VR glasses, but there are still limitations. Lemsalu et al. (2022) used YOLOv5 to detect strawberry fruits and applied it to a strawberry harvesting robot, but they also explicitly pointed out that the model could not correctly detect densely clustered fruits. Zhang et al. (2022) improved the lightweight YOLOv4-tiny model, mainly for the harvest detection of ripe strawberries. Although it maintained high detection accuracy and speed, it depended on high-configuration Jetson Nano hardware, limiting the model’s application on low-power devices. In summary, although some research has been made on strawberry detection, applying the detection and classification of strawberry fruits to harvesting robots has solved the problem of automated strawberry harvesting to some extent. However, with the increase in the number of layers and model parameters of deep learning networks, some networks have become very complex and large, requiring high computational resources, so there is still room for improving efficiency.

In addition, strawberry fruits growing naturally in the environment are densely distributed, present in various postures, and often overlap and grow in the shade. Delbridge (2021) analyzed the current state of robotic strawberry harvesting. Although this is an important direction for the future development of the strawberry industry, to achieve economic feasibility, it is necessary to further improve technology and algorithm levels.

In response to these challenges, we propose a structurally simple and efficient object detection model: the Strawberry Lightweight Feature Classify Network (SLFCNet). This model is based on the lightweight backbone network of Cross Stage Partial Network (CSPDarknet) (Bochkovskiy, Wang & Liao, 2020) and applies the Combined Convolutional Concatenation and Sequential Convolutional (C3SC) module for feature decoding, effectively integrating the features of multi-scale targets. SLFCNet not only maintains a small number of parameters but also significantly improves detection performance, making the model size only 3.57 MB, and the inference time for each image is only 4.1 ms on the GTX 1080 Ti GPU, which is very suitable for deployment on mobile and embedded devices with limited computing power. This promotes the popularization of automated strawberry harvesting, thereby improving the production efficiency and quality of strawberries in a targeted manner.

In summary, our main contributions are threefold:

(1) We propose the C3SC module, which simplifies the structure while enhancing efficiency, reducing the parameter count in the neural network’s decoder, and improving feature extraction capabilities.

(2) We introduce the lightweight SLFCNet model, suitable for real-time detection and classification of strawberries at various growth stages in the field, which can be further deployed on edge devices, offering a novel solution for automated strawberry harvesting.

(3) We compare the SLFCNet model with mainstream object detection models, demonstrating its superior robustness and adaptability in complex environments.

Materials and Methods

Dataset

The dataset utilized in this study originates from the publicly available Strawberry-DS dataset (Nergiz, 2023; Elhariri, El-Bendary & Saleh, 2023), which includes 247 high-resolution images captured in a greenhouse at the Agricultural Research Center in Giza, Egypt, with a resolution of 3,840 × 2,160. These images depict the transformation of strawberries through various stages of ripening from “Green” to “Red”, with the number of strawberries in each image ranging from 1 to 25. To ensure data accuracy, the bounding box annotations were checked and corrected using the Labelimg tool (Tzutalin, 2022). The fruits are categorized into six distinct growth stages using different labels: “Green”, “White”, “Early-turning”, “Turning”, “Late-turning”, and “Red”. The dataset images include strawberries that are either fully visible or partially occluded, presenting a challenging task for object detection and classification models to accurately identify these strawberries for automated harvesting. As demonstrated in Fig. 1, typical cases have been selected from the dataset for explanation.

Figure 1 Four main challenges of strawberry classification task (A–D).

Image and Data source credit: El-Bendary & Elhariri (2022).

In Figs. 1A1, 1A2, and 1A3, occluded by leaves or branches: Strawberry fruits are obscured by leaves or branches, sometimes making them difficult to detect even by the human eye. In Fig. 1B, mutual occlusion between different fruits: Densely arranged strawberry fruits overlap visually, interfering with the model’s recognition. In Fig. 1C, abnormal growth shapes: Some fruits exhibit abnormal shapes due to growing environments or external forces. In Fig. 1D, Small individual size: Underdeveloped small strawberries or newly sprouted strawberries, wrapped by the calyx (Hervieux et al., 2016), require the model to have high resolution and sensitivity for accurate detection and classification.

In this study, the Strawberry-DS dataset primarily consists of images of strawberries captured in a greenhouse environment, which limits the diversity of the dataset and the simulation of real-world scenarios. To overcome this limitation, various image enhancement techniques were employed to augment and diversify the dataset. These techniques include adjusting image brightness and contrast, applying multi-angle rotations and flips, and adding noise. These operations not only increase the number of images but also simulate different environmental conditions, such as varying lighting and weather, thereby improving the model’s generalization and robustness. The specific parameter settings are as follows: (1) Image brightness is randomly adjusted between 0.5 and 1.5 times to adapt to different environmental lighting; (2) image contrast is randomly adjusted between 0.5 and 1.5 times to reveal different image details; (3) random multi-angle rotations are performed, such as 45°, 90°, 270°, horizontal flips, and mirror flips. With these enhancement measures, the total size of the dataset is significantly expanded, increasing the diversity of perspectives and postures of the strawberry fruits. The enhanced dataset is divided into training, validation, and testing sets in an approximate ratio of 8:1:1 to support model training and evaluation. These measures help improve the model’s performance in practical applications, especially in scenarios of automated strawberry harvesting. Detailed information about the dataset is shown in Table 1.

Table 1 Detailed information of the dataset.

Dataset	Original image	Augmented image	Augmented label	
			Total	Green	White	Early- turning	Turning	Late- turning	Red	
Total	247	1,485	6,498	2,628	1,504	324	289	363	1,390	
Training	172	1,235	5,417	2,141	1,227	284	252	298	1,215	
Validation	49	125	592	276	134	21	28	38	95	
Test	26	125	489	211	143	19	9	27	80	

Model architecture

In this work, considering the specific requirements of the task, the lightweight backbone network CSPDarknet (Bochkovskiy, Wang & Liao, 2020) was selected as the foundation for feature extraction. This network is renowned for its excellent feature extraction capabilities and rapid detection speed, enabling an increase in processing speed without sacrificing accuracy. Figure 2 illustrates the network structure of SLFCNet, and we will now provide a detailed explanation of each component.

Figure 2 Architecture of SLFCNet.

Image and data source credit: El-Bendary & Elhariri (2022).

Encoder

The role of the encoder is to map the input RGB image into feature maps, which will undergo further processing for strawberry detection and classification. Within the encoder, an RGB image with dimensions H ×W ×3 is input into the model, defined as I ∈ RH×W×C, where H represents the image’s height, W its width, and C the number of feature channels. After encoding, the feature maps undergo feature extraction. We employ CSPDarknet as the backbone feature extraction part because it effectively enhances feature extraction capability while maintaining model lightweight through the design of Cross Stage Partial Connections. The entire Encoder process comprises 5 convolutional layers and 4 feature extraction layers. Features are extracted at different stages through the insertion of CSPLayer (Wang et al., 2020a; Wang et al., 2020b), resulting in output channel counts of 32, 64, 128, and 256, generating feature maps at different stages. It is specifically structured as C3(16)-C3(32)-M(32)-C3(64)-M(64)-C3(128)-M(128)-C3(256)-M(256)-S5(256), where Ck(m) denotes 2D convolution layers BSConv with m channels and k × k filters, with a stride of 2, composed of Conv2d + BatchNorm2d + SiLU activation function (Elfwing, Uchibe & Doya, 2018), structured as illustrated in Fig. 2A. M(k) represents a feature extraction layer C2f with an output of k channels, structured as illustrated in Fig. 2B, utilizing three convolution modules and n BottleNeck structures for feature fusion and propagation. This design allows the C2f module to effectively fuse and propagate features at different levels, enhancing the network’s feature representation. The final S5(256) with an input channel count Cinput = 256 and a kernel size KSPPF = 5 × 5 is the SPPF (Spatial Pyramid pooling - fast) module (Jocher, 2020), as illustrated in Fig. 2C. It concatenates images from max-pooling operations, improving feature utilization efficiency and reducing network parameters. Ultimately, this reduces the feature map to 1/32 of the original image size.

Decoder

Previous experiments and experience has emphasized the significance of leveraging information from diverse feature levels (Yu, Ye & Zhou, 2023; Ye et al., 2023; Lu et al., 2023). The decoder fulfills a pivotal role in combining and decoding the predictive information extracted by the encoder, ensuring the effective utilization of the multi-level features. After experimentation, we decided to incorporate an SA attention module for feature reweighting and adjustment, to be validated in subsequent steps. This SA attention module is applied at the interface between the encoder and decoder. It divides the input feature tensor into multiple channel groups and randomly permutes the channels within each group. After dimension reduction through BSConv, we believe that at this point, this low-level feature layer is ready for predictions in the Head section, as it can provide richer positional information. This connection strategy aligns with the principles expressed by feature pyramid networks (FPN) (Lin et al., 2017).

Next, the decoder part was innovatively designed with an optimized connection strategy. Considering the need to better obtain high-level features, increase semantic and category information, and maintain the model’s lightweight nature, the C3SC (Combined Convolutional Concatenation and Sequential Convolutional) module was designed, as shown in Fig. 2D. This module employs nearest neighbor interpolation on the upper-level features to double the feature map size. Subsequently, we perform a Concat operation with feature maps of the same size from the third-stage output of the decoder. However, after this connection, we do not rush to pass it to the head for detection. Instead, we further connect it to a standard 1 × 1 convolution layer (BSConv) to reduce the model’s parameter count and computational complexity. This set of features is then incorporated as the second branch of the decoder and forwarded to the head. To utilize feature information from different scales, we devise a concatenated structure of multiple C3SC modules. Each C3SC module takes the output features from the previous stage as input and undergoes the same operations. Through iterative up-sampling and feature fusion, we obtain three sets of feature maps at different scales, enhancing the model’s perception of objects of various sizes, which are utilized for prediction.

In summary, in the design of the decoder, SLFCNet prioritizes maintaining a lightweight model and optimizing feature fusion strategies to avoid errors in positional information from deep network layers. Through multiple up-sampling iterations to integrate semantic information while timely propagating low and high-level features to the Head, we ensure that the fusion effect of non-adjacent levels is not compromised. This design not only reduces feature loss and degradation but also enhances the model’s ability to predict the position and category of strawberry fruits. Compared to other models, SLFCNet’s Decoder structure is more refined and efficient, making it highly suitable for deployment in real-world scenarios.

Loss function

During the construction of neural network models, the choice of the loss function is pivotal for the model’s performance. In the task of strawberry detection, we specifically focus on two commonly used loss functions: CIoU (Zheng et al., 2021) and SIoU (Gevorgyan, 2022).

Firstly, both CIoU and SIoU incorporate Intersection over Union (IoU) as a crucial component in their respective frameworks. IoU is employed to measure the overlap between the predicted bounding box and the ground truth bounding box, defined as the intersection area of two bounding boxes divided by their union area. This value serves as an indicator of both the accuracy and recall associated with the predicted box, which is often used to evaluate the performance of object detection algorithms. The calculation formula is presented in Eq. (1): (1) IoU=B∩BGTB∪BGT

where B represents the predicted box, and BGT represents the ground truth box. A higher IoU value signifies a greater overlap between the two boxes. In the CIoU loss function, IoU is utilized to compute the initial intersection over union. Then, based on the distance between the predicted box and the ground truth box, as well as their aspect ratios, IoU is adjusted to obtain the final CIoU value. The calculation formula is shown in Eq. (2): (2) CIoU=IoU−ρ2b,bGTc2−αv

where ρ2b,bGT represents the square of the Euclidean distance between the centers of the predicted box and the ground truth box, c represents the length of the diagonal of the minimum bounding box of the predicted box and the ground truth box. v represents the consistency of the aspect ratio between the predicted box and the ground truth box, and the calculation formula is shown in Eq. (3): (3) v=4π2arctanwGThGT−arctanwh2

where w and h represent the width and height of the predicted box, wGT and hGT represent the width and height of the ground truth box. A smaller value of v indicates a closer aspect ratio between the predicted box and the ground truth box.

α is a positive balancing parameter used to adjust the weight of v in the loss function, and the calculation formula is shown in Eq. (4): (4) α=v1−IoU+v

The CIoU loss function comprehensively evaluates the accuracy of the predicted box by considering IoU, center point distance, and aspect ratio consistency. However, a major drawback of CIoU is that it fails to consider the directional difference between the target box and the predicted box, which may result in a slower convergence speed for the model in some complex scenarios. To overcome this limitation, the SIoU loss function was proposed and widely applied. SIoU introduces the vector angle between the target box and the predicted box, building upon CIoU, as shown in Eq. (5): (5) LSIoU=1−IoU+Δ+Ω2

where Δ represents the angle cost, which is utilized to measure the directional difference between the predicted box and the ground truth box. Ω represents the shape cost, serving to evaluate the similarity in shape between the predicted box and the ground truth box. The calculation formula for the angle cost Δ is presented in Eq. (6), whereas the calculation formula for the shape cost Ω is detailed in Eq. (7): (6) Δ= ∑t=x,y1−e−γρt

(7) Ω= ∑t=w,h1−e−wtθ

Here, wt and ht respectively represent the relative differences in width and height between the predicted box and the ground truth box. θ is a parameter of focus degree used to adjust the weight of the shape cost in the loss function. The cost value is inversely proportional to the shape difference, obtained by subtracting the exponential function’s value from 1.

The SIoU loss function provides a more comprehensive assessment of the differences between the predicted and actual bounding boxes by taking into account both the angular and shape costs. It not only focuses on the positional relationship but also enhances the accuracy of the predicted bounding box. In the task of strawberry detection, the use of the SIoU loss function can speed up the convergence of the network and improve the precision of the regression, making the predicted target frames more stable and the detection results closer to the real targets.

Results

Experiment details

In this study, we utilized a total of 1,485 images from the augmented Strawberry-DS dataset, randomly partitioned into training, validation, and test sets in an 8:1:1 ratio. The training set contained 1,235 images, while both the validation and test sets contained 125 images each. The specific label distribution is provided in Table 1. The experiments were conducted using version 2.0.0 of the PyTorch deep learning framework (Paszke et al., 2019), utilizing CUDA 11.8 for parallel computation capabilities to accelerate both model training and inference processes. Additionally, PyTorch offered a rich set of pre-trained models, optimization algorithms, and visualization tools, which greatly aided the development and evaluation of SLFCNet. Taking into account the high resolution of the samples, we resized the input images to 640 ×640 pixels, aligning well with the requirements of edge devices. To ensure the objectivity of the results, all methods were trained and tested under the same configuration. During training, considering the dataset size and model complexity, we conducted 450 epochs with a batch size of 16. The learning rate was initialized to 0.01, and we employed the stochastic gradient descent (SGD) optimizer. SGD computes gradients and updates model parameters utilizing only a subset of samples in each iteration, thereby avoiding wasting computational resources. We set the momentum factor to 0.937 to strike a balance between faster convergence and reduced oscillations during training for stability. The weight decay was set to 5 × 10−4 to encourage the model to select smaller weight values during training, thereby enhancing generalization. To prevent overfitting and improve model robustness, we applied data augmentation strategies to expand the dataset and simulate more complex environmental conditions.

Assessment metrics

In our experimental setup, we employed widely adopted performance metrics, including Precision (P), Recall (R), F1-score (F1), mean average precision (mAP), and optimal localization recall precision (oLRP) (Oksuz et al., 2018), to assess the detection results for strawberries. Precision (P) serves as an evaluation of the model’s predictive outcomes, quantifying the proportion of true positive samples among those predicted as positive. Meanwhile, recall (R) assesses the true samples, indicating the fraction of actual positive samples correctly predicted. The computation methods for P and R are delineated by Eqs. (8) and (9): (8) P=TPTP+FP

(9) R=TPTP+FN

where true positives (TP) represent the number of correctly identified positive samples, false positives (FP) signify the number of samples erroneously classified as positive, and false negatives (FN) indicate the number of samples incorrectly recognized as negative within the positive class. “TP + FP” represents the total count of detected strawberries, and “TP + FN” represents the total count of true strawberries in the image.

The metric mAP denotes the comprehensive performance at various Intersection over Union (IoU) thresholds, encompassing mAP@0.5 and mAP@0.5:0.95. The F1-score, achieved by striking a balance between precision and recall, serves as a performance assessment tool, affording equitable consideration to the P and R metrics. oLRP denotes the minimum achievable LRP error at τ = 0.5. The computational expressions for these metrics are presented as Eqs. (10) to (13): (10) mAP=1n∑1nPRdR

(11) F1=2×P×RP+R

(12) LRPX,Ys=1NTP+NFP+NFN∑i=1NTP1−IoUxi,yxi1−τ+NFP+NFN

(13) oLRP=minsLRPX,YS

Here, n represents the number of categories. In this experiment, strawberries are categorized into six distinct growth stages, making n = 6. mAP@0.5 signifies the mean average precision at an IoU threshold of 0.5. A higher value indicates greater detection precision for that category. mAP@0.5:0.95, on the other hand, signifies the mean average precision across various IoU thresholds, ranging from 0.5 to 0.95 with a step size of 0.05. This metric imposes stricter performance requirements on the model. Conversely, the oLRP value represents the error rate, and typically, a lower value signifies better performance of the model in terms of target localization.

Analysis of detection results by SLFCNet

To visually present the changes in loss and performance over the number of epochs, we plotted a comprehensive view in Fig. 3. It displays the box_loss (Terven & Cordova-Esparza, 2023), cls_loss, and Distribution Focal Loss (dfl_loss) during the training and validation processes. The results indicate that these loss values and the mAP metric stabilize after approximately 150 epochs, with no signs of overfitting, demonstrating the appropriateness of the training parameters.

Figure 3 The loss and performance results along the epoch number for training and validation datasets.

Furthermore, we plotted the Precision, Recall, Precision × Recall, and F1-score curves for the SLFCNet model on the test dataset, as shown in Fig. 4. Meanwhile, Table 2 lists the detailed performance metrics for all categories. It is observed that as the confidence threshold increases, Precision improves while Recall decreases, reflecting the contradictory relationship between the two. Notably, when the confidence level is around 0.7, the performance of all categories reaches a balance.

Figure 4 The precision, recall, precision × recall and F1-score curves of test dataset.

Our analysis suggests that the detection performance for the “Green” and “White” categories of strawberries is relatively poor, mainly for two reasons: firstly, the color of these two ripening stages is similar to the background, which can easily cause confusion in identification; secondly, the volume of these stage strawberries is smaller, especially green strawberries, which are often wrapped by the calyx, leading to easy miss-detection or false detection. Another interesting observation is that as the strawberries ripen, the “Red” category, due to its distinct color characteristics, achieves the best performance, showing higher mAP and F1-scores, as it has more prominent color features.

Table 2 Performance results of validation dataset for each class.

Class	Images	P	R	F1	mAP@0.5	mAP@0.5:0.95	
Total	125	0.947	0.932	0.939	0.989	0.772	
Green	125	0.977	0.733	0.838	0.966	0.561	
White	125	0.992	0.857	0.920	0.988	0.742	
Early-turning	125	0.873	1.000	0.932	0.995	0.796	
Turning	125	0.957	1.000	0.978	0.995	0.823	
Late-turning	125	0.886	1.000	0.940	0.995	0.895	
Red	125	0.995	1.000	0.997	0.995	0.813	

Additionally, Table 2 also reflects that in the “Early-turning”, “Turning”, “Late-turning”, and “Red” categories, the recall (R) has reached 1, and the mAP@0.5 and mAP@0.5:0.95 performances are also good. This may be related to the small number of samples of these categories in the dataset, making any performance changes have a significant impact on the results. Nevertheless, the SLFCNet model successfully identified and classified these strawberries through training, demonstrating good generalization and robustness.

Comparative experiments with other models

To evaluate the performance of the SLFCNet model, we set up comparative experiments between the SLFCNet model and four mainstream object detection models, including YOLOv5 (Jocher, 2020), YOLOv8 (Jocher, Chaurasia & Qiu, 2023), Faster R-CNN (Ren et al., 2016) and SSD (Liu et al., 2016). These models are representative and have excellent performance in the field of object detection, where YOLOv5 and YOLOv8 are single-stage models, Faster R-CNN is a two-stage model, and SSD focuses on multi-scale detection. All models were trained and evaluated under the same conditions, using an image resolution of 640 × 640 pixels.

The experimental results from Table 3 show significant differences in performance among different models. YOLOv8 performed well in F1-scores and mAP@0.5, while Faster R-CNN and SSD had advantages in inference time, but SSD’s F1-score was lower. SLFCNet surpassed other models in all evaluation metrics, especially with an inference time of only 4.1 ms, significantly faster than YOLOv8 and other models.

Table 3 Comparison of counting performance for different model.

The best results are indicated in bold black.

Models	P	R	F1	mAP@0.5	mAP@0.5:0.95	oLRP	Inference time	
YOLOv5	0.762	0.872	0.813	0.866	0.694	0.24	6.6 ms	
YOLOv8	0.937	0.907	0.922	0.965	0.765	0.15	5.8 ms	
SSD	0.579	0.758	0.657	0.727	0.618	0.35	5.8 ms	
Faster R-CNN	0.798	0.853	0.825	0.869	0.658	0.22	10.6 ms	
SLFCNet	0.947	0.932	0.939	0.989	0.772	0.10	4.1 ms	

Further analysis found that the models’ performance in handling small targets, occlusions, and background confusion varied. YOLOv8 and SLFCNet had better detection capabilities for occluded strawberries, while SSD and Faster R-CNN performed poorly. In addition, SLFCNet showed better robustness against background interference, accurately identifying targets in complex environments. In summary, SLFCNet demonstrated excellent performance in strawberry detection tasks, especially in detection efficiency and accuracy, meeting the real-time detection requirements for automated harvesting robots.

To identify the causes affecting detection performance, we selected four representative images to analyze the performance of the five models in actual testing, mainly observing missed and false detection situations. As shown in Fig. 5, the analysis results are as follows:

Figure 5 Comparison of the effect of different detection algorithms.

(A) YOLOv5; (B) YOLOv8; (C) SSD; (D) Faster R-CNN; (E) SLFCNet. Blue round boxes indicate cases of missed detections, and yellow round boxes represent cases of false positives. Image and data source credit: El-Bendary & Elhariri (2022).

(1) Individual size: In the first image, the strawberries are relatively small, and both the YOLOv8 and Faster R-CNN models exhibit missed detections, while the SSD model fails to detect the strawberries entirely, indicating its relatively poor robustness in handling small objects. This trend is further validated in subsequent images. (2) Occlusion Interference: In the second and third images, strawberries are partially occluded and overlapped to varying degrees. In such cases, YOLOv8 performed relatively well, followed by YOLOv5, but SSD and Faster R-CNN models could hardly detect these strawberries. (3) Background Confusion: In the upper left corner of the fourth image, there are two small strawberries. When the background is similar in color to the target or strawberries are occluded by the calyx, most models struggle to correctly identify strawberries. Although YOLOv8 could identify some strawberries with low confidence, and SLFCNet identified the targets but also produced some false detections.

These challenges are common in automated strawberry harvesting, and their main sources of error are the occlusion of the target of interest by leaves or other parts of the canopy, leading to inaccuracies in the detection process. In addition, occasionally, objects with colors and shapes similar to strawberries appear in the clutter, which can also lead to incorrect detections. SLFCNet showed good robustness in these tests, effectively handling small targets, occlusions, and background interference, improving the recognition rate and confidence, demonstrating its advantages in practical applications.

Attention mechanism ablation experiment

The results presented above clearly indicate that the designed SLFCNet in this study achieves the most accurate detection outcomes for strawberries at different growth stages, confirming the efficacy of the model. As previously mentioned, SLFCNet incorporates the Shuffle Attention (SA) module (Zhang & Yang, 2021), alongside other widely attention modules such as the Squeeze-and-Excitation Attention Module (SE) (Hu, Shen & Sun, 2018), Convolutional Block Attention Module (CBAM) (Woo et al., 2018), and Efficient Channel Attention (ECA) (Wang et al., 2020a; Wang et al., 2020b). To further evaluate the effectiveness of these attention mechanisms in strawberry fruit detection at different growth stages, we conducted ablation experiments, evaluated them using standard metrics, and compared the parameter count for each model. The findings are summarized in Table 4.

Table 4 Impact of different attention mechanisms on the mode.

In the “Models” column, “base” represents the SLFCNet network model without any attention mechanism, “+” denotes the addition of the respective module on the “base” model, The best results are indicated in bold black.

Models	P	R	F1	mAP@0.5	mAP@0.5:0.95	Inference time	Parameters	
base	0.891	0.921	0.906	0.972	0.767	4.4 ms	179,2434	
+SE	0.917	0.897	0.907	0.971	0.784	4.3 ms	180,0626	
+CBAM	0.890	0.905	0.897	0.969	0.788	4.3 ms	185,8324	
+ECA	0.908	0.896	0.902	0.977	0.760	4.2 ms	179,2437	
+SA	0.947	0.932	0.939	0.989	0.772	4.1 ms	179,2530	

The experimental outcomes indicate that the approach utilizing the SA attention mechanism achieves superior detection performance and inference efficiency. In detection tasks, the SE attention mechanism only considers attention in the channel dimension, neglecting to capture attention in the spatial dimension and thereby disregarding crucial positional information and spatial structure. The CBAM attention mechanism is suitable for scenarios that require effective integration of both spatial and channel dimensions in feature maps. However, it sacrifices attention in both dimensions, demanding higher computational resources and resulting in increased computational complexity. The ECA attention mechanism, while relatively efficient and introducing only a small number of parameters, exhibits limitations in handling global context dependencies and channel-spatial relationships, thus falling short of optimal performance. Taking all factors into account, we have chosen to integrate an SA attention module into the SLFCNet architecture, in conjunction with the C3SC module, to enhance the model’s global feature extraction and spatial interaction capabilities, and to recalibrate feature weighting. This strategic choice ensures both high detection accuracy and efficient utilization of computational resources, contributing to the overall effectiveness of the SLFCNet model in strawberry fruit detection tasks.

Data augmentation ablation experiment

In deep learning, data augmentation is a commonly employed technique. Its aim is to enhance a model’s generalization ability and mitigate overfitting by transforming or augmenting the original dataset. In this study, to quantitatively assess the impact of data augmentation strategies, we conducted an ablation experiment and compared the performance changes of different models before and after applying data augmentation.

Firstly, let us review the performance of the SLFCNet model on the augmented dataset, as depicted in Table 2. From the table, it is evident that SLFCNet achieved satisfactory results across multiple evaluation metrics. However, it is noteworthy that we observed some performance discrepancies among different categories, specifically lower precision rates for the “Early-turning” and “Late-turning” categories. Next, the performance of the SLFCNet model on the original dataset without any data augmentation strategy was recorded as shown in Table 5.

Table 5 Comparison of SLFCNet performance on the original dataset.

Class	P	R	F1	mAP@0.5	mAP@0.5:0.95	
Total	0.704	0.650	0.676	0.688	0.510	
Green	0.750	0.702	0.725	0.759	0.466	
White	0.648	0.681	0.664	0.595	0.414	
Early-turning	0.556	0.714	0.626	0.688	0.595	
Turning	0.825	0.500	0.623	0.677	0.498	
Late-turning	0.535	0.357	0.428	0.488	0.338	
Red	0.912	0.944	0.928	0.920	0.745	

Additionally, to comprehensively evaluate the effectiveness of data augmentation strategies, we selected several popular object detection models for comparison, including YOLOv5, YOLOv8, SSD, and Faster R-CNN. Similarly, Table 3 presents their performance on the augmented dataset, whereas Table 6 displays their performance without the employment of data augmentation strategies. It can be observed that, prior to the adoption of data augmentation strategies, all models exhibited relatively high oLRP values, indicating a significant loss in localization accuracy and suggesting that there is room for improvement in localization performance. Especially for the SSD model, its oLRP value reached 0.61, showing a more pronounced localization accuracy issue. The architectural design of the model has a significant impact on performance improvement. For example, Faster R-CNN adopts a two-stage detection approach, combining a Region Proposal Network (RPN) (Ren et al., 2016) with a classification and regression network, enabling it to better extract key features from complex data and significantly enhance performance after data augmentation. The YOLOv8 and SLFCNet models also have strong feature extraction and adaptability capabilities, thus they also performed well and showed a significant performance improvement after data augmentation.

Table 6 Comparison of performance of different models on the original dataset.

Models	P	R	F1	mAP@0.5	mAP@0.5:0.95	oLRP	
YOLOv5	0.695	0.704	0.699	0.735	0.544	0.36	
YOLOv8	0.677	0.654	0.665	0.681	0.509	0.43	
SSD	0.369	0.622	0.463	0.546	0.356	0.61	
Faster R-CNN	0.802	0.668	0.729	0.599	0.406	0.40	
SLFCNet	0.704	0.650	0.676	0.688	0.510	0.45	

Ablation study of different parameter settings

In the experiments with the SLFCNet model, we thoroughly investigated the specific impact of different hyperparameter settings on model performance. Specifically, we conducted extensive experiments on the validation set by adjusting key hyperparameters such as the number of iterations (epochs), batch size, learning rate, and optimizer. Table 7 summarizes the results of these experiments, presenting a comparison of model performance under different settings.

Table 7 Comparison of model performance under different hyperparameter settings.

Epochs	Batch size	Learning rate	Optimizer	Validation accuracy	
300	16	0.01	SGD	0.875	
450	16	0.01	SGD	0.989	
600	16	0.01	SGD	0.981	
450	8	0.01	SGD	0.978	
450	32	0.01	SGD	0.987	
450	16	0.001	SGD	0.936	
450	16	0.1	SGD	0.915	
450	16	0.01	Adam	0.965	
450	16	0.01	Adagrad	0.959	

The experimental results indicate that 450 epochs of iteration yield the optimal model performance, with a validation accuracy of 0.989, which is a significant improvement over the 0.875 accuracy at 300 epochs. A slight decrease in accuracy to 0.981 was observed at 600 epochs, which may suggest the initial signs of overfitting. Adjustments to the batch size revealed that a setting of 16 outperformed smaller or larger batch sizes (8 and 32). Sensitivity analysis of the learning rate identified 0.01 as the optimal choice, with learning rates of 0.001 and 0.1 leading to performance drops to 0.936 and 0.915, respectively. Furthermore, the choice of optimizer was equally crucial for performance, with the Stochastic Gradient Descent (SGD) optimizer outperforming Adam (Kingma & Ba, 2014) and Adagrad (Duchi, Hazan & Singer, 2011) under these experimental conditions, with the latter two achieving accuracies of 0.965 and 0.959, respectively.

Model feature visualization analysis

In this study, to enhance the interpretability of the model, we employed Gradient-weighted Class Activation Mapping (Grad-CAM) (Selvaraju et al., 2017) technology for a visual analysis of the output layer of the SLFCNet model. With this technique, we were able to demonstrate the model’s attention to specific areas when processing images containing strawberries with occlusions, overlaps, and varying sizes. These areas are represented by different colors ranging from blue to red, with red indicating the highest level of attention. As depicted in Fig. 6, the SLFCNet model encounters interference from the field environment, such as foliage and debris, resulting in the attention area encompassing both strawberry fruit characteristics and some background and texture information. While this interference may impact the model’s recognition results to some extent, it is crucial to note that the SLFCNet model varies its level of attention to each individual strawberry, maintaining reliable overall recognition performance. It adeptly focuses on the distinctive features of strawberries at different growth stages, efficiently filters out extraneous background information, while accentuating the target fruit, thereby emphasizing its superior performance.

Figure 6 Visualization analysis.

From blue to red, it indicates that the model pays increasing attention to the region. Image and data source credit: El-Bendary & Elhariri (2022).

Discussion

This article presents the SLFCNet model, which is supported by a lightweight CSPDarknet backbone and introduces a simple and efficient C3SC module in the decoder to effectively fuse multi-scale targets. It is worth mentioning that the SLFCNet model not only has high performance but also has a small size of only 3.57 MB. This makes it highly suitable for deployment on small harvesting robots, reducing the requirements for hardware equipment and thereby decreasing the deployment costs for agricultural managers. At the same time, by adopting a series of image enhancement strategies, the model’s robustness and generalization in complex backgrounds and occlusion interference have been significantly improved.

The motivation for this study stems from considerations of the issues present in automated strawberry harvesting, including lightweight requirements, real-time processing, inference speed, and hardware resource demands. Although the SLFCNet has performed well in many aspects, it also has limitations, including the complexity of actual field images, occlusion interference, and the detection difficulties posed by strawberries at different growth stages. To address these challenges, we have adopted data augmentation strategies that expose the model to a wider variety of occlusion scenarios during training, thereby enhancing the model’s robustness.

Conclusions

In conclusion, this study presents SLFCNet, a strawberry fruit growth stage detection method that integrates the concise and efficient Combined Convolutional Concatenation and Sequential Convolutional (C3SC) module, ensuring lightweight and rapid inference efficiency. SLFCNet demonstrates its ability to precisely and swiftly detect and classify strawberry fruits across six distinct growth stages in real-time.

SLFCNet is particularly well-suited for low-end edge devices, such as harvesting robots and video surveillance equipment, as well as existing harvesting platforms. SLFCNet holds significant potential in supporting robot strawberry harvesting. It promises to enhance harvesting speed and efficiency, minimize human intervention, and assist growers in promptly identifying and harvesting ripe strawberries. Consequently, this could maximize economic benefits for growers.

In future work, we intend to enhance the diversity of the strawberry dataset by capturing images under various lighting and weather conditions, with a specific focus on collecting more images taken at night. This expansion aims to enhance the robustness of the object detection model. Additionally, we aim to assess the model’s performance on specific embedded devices and further optimize it, contributing to the advancement of unmanned farms and precision agriculture.

Supplemental Information

Supplemental Information 1 SLFCNet Code

Additional Information and Declarations

Competing Interests

Author Contributions

Data Availability

The authors declare there are no competing interests.

Wenchao Xu conceived and designed the experiments, analyzed the data, performed the computation work, prepared figures and/or tables, authored or reviewed drafts of the article, and approved the final draft.

Yangxu Wang conceived and designed the experiments, performed the experiments, analyzed the data, performed the computation work, prepared figures and/or tables, authored or reviewed drafts of the article, and approved the final draft.

Jiahao Yang performed the experiments, analyzed the data, prepared figures and/or tables, authored or reviewed drafts of the article, and approved the final draft.

The following information was supplied regarding data availability:

The data is available at GitHub and Mendeley Data:

- https://github.com/YangxuWangamI/SLFCNet.

- El-Bendary, Nashwa; Elhariri, Esraa (2022), “Strawberry-DS”, Mendeley Data, V1, doi: 10.17632/z6dtfdpzz8.1. Available at https://data.mendeley.com/datasets/z6dtfdpzz8/1.

The datasets used in this study are available at figshare: Wang, Yangxu (2024). Untitled Item. figshare. Dataset. https://doi.org/10.6084/m9.figshare.25686882.v1.

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
