# Peer review of "SLFCNet: an ultra-lightweight and efficient strawberry feature classification network"

_PeerJ Computer Science, doi:10.7717/peerj-cs.2085_

## Round 0.1 · original submission · Major Revisions

Dear authors,

You are advised to critically respond to all comments point by point when preparing a new version of the manuscript and while preparing for the rebuttal letter. Please address all the comments/suggestions provided by the reviewers.

Kind regards,
PCoelho

**Language Note:** PeerJ staff have identified that the English language needs to be improved. When you prepare your next revision, please either (i) have a colleague who is proficient in English and familiar with the subject matter review your manuscript, or (ii) contact a professional editing service to review your manuscript. PeerJ can provide language editing services - you can contact us at [email protected] for pricing (be sure to provide your manuscript number and title). – PeerJ Staff

Reviewer 1 ·

Basic reporting

I appreciate how the author presents their work in this paper. The manuscript is written professionally.
However, some of the terms presented still require an explanation and meaning of the abbreviations presented such as CBAM, CSP, C3HB, and so on.

Experimental design

In Introduction (page 6): Attention Mechanism. Should "CBAM" be "BCAM"?

In Results
page 13
This dataset has increased in terms of quantity due to the augmentation process. It would be better to add an explanation of how much data grows for each augmentation technique that was explained in the previous dataset section.

page 19:
The using of occlusion and augmentation has shown an increased number of datasets. Also, in this discussion mentioned that the various occlusion given a problem in strawberry detection. The statement should be explained clearly about the reasons and what type of occlusion which proposed the issues.

Validity of the findings

no comment

Additional comments

All comments should be revised upon before Acceptance.

Annotated reviews are not available for download in order to protect the identity of reviewers who chose to remain anonymous.

Reviewer 2 ·

Basic reporting

no comment

Experimental design

no comment

Validity of the findings

Could have been better. Dataset with only 247 images is too small to analyze the data.

Additional comments

The paper will be readable if the context is justified.

Reviewer 3 ·

Basic reporting

All comments have been added in detail to the 4th section called additional comments.

Experimental design

All comments have been added in detail to the 4th section called additional comments.

Validity of the findings

All comments have been added in detail to the 4th section called additional comments.

Additional comments

Review Report for PeerJ Computer Science
(SLFCNet: an ultra-lightweight and efficient strawberry feature classification network)

1. Within the scope of the study, various deep learning models were used to classification and detect strawberry fruits and a model specific to the study was created.

2. Although there are many deep learning-based models for object detection in the literature, it should be explained in more detail how the other models compared with the model proposed in this study were selected. Interpret it in terms of single-stage and two-stage object detection models.

3. Open source datasets were used for Strawberry data. It is important to use data augmentation instead of using this 6-class dataset as raw. The results obtained in classification and object detection problems are very dependent on the dataset. For this reason, it is very important how the test dataset is selected. It was stated in the study that the training, validation and test datasets were selected in a ratio of 8:1:1. How were the images in the test dataset selected? If the images in the test dataset were changed, would there be positive/negative effects on the classification/detection results? Why wasn't cross-validation used instead of random selection of the dataset distribution?

4. It is important to perform data augmentation within the scope of the study. But how the effect of data augmentation on classification/detection results should be explained in more detail? Have the results improved before and after data augmentation, or what kind of improvement has occurred?

5. For correct analysis and interpretation of the results, evaluation metrics must be obtained completely. For this reason, for all recommended and compared models, with and without augmentation, please add the missing evaluation metrics such as Average Recall (AR), Optimal Localization Recall Precision (oLRP), count of predicted bounding box.

6. Each part of the architecture given in Figure 2 can be explained in more detail, including the proposed algorithm step by step. It should be explained why the blocks were selected, how it differs from the literature, and why CSPDarknet was chosen even though there are many different models that can be used in the literature, such as backbone feature extraction. Should the contribution of originality here be expressed more clearly?

7. It is recommended that the data amounts for the initial (non-augmented) and augmented states (with training, validation, test data) for each class in the dataset be given in a table and explained in more detail.

8. For the experiment part, should the framework/toolbox etc. used within the scope of the study be explained in more detail?

9. It should be explained how the hyperparameters used in the training phase were selected. Why epoch 450? How does having less or more affect the results? How was it determined? Why is the batch size value 16? It should be explained how the learning rate value, optimizer, etc., all other parameters were selected, and whether other experiments were carried out?

As a result, although the study is of a certain quality, it is recommended that all the sections mentioned above be taken into consideration in order to reveal its contribution to the literature and its originality more clearly.

---

## Round 0.2 · Major Revisions

Dear authors,

The reviewers are not totally satisfied with the provided manuscript after the previous round. It is stated that you require to address ALL the comments point by point when preparing a new version of the manuscript and also prepare a rebuttal letter. Re-check the previous round comments and provide the ones that are missing. Please address all the comments/suggestions provided by the reviewers.

Kind regards,

PCoelho

Reviewer 2 ·

Basic reporting

no comment

Experimental design

Research gaps could be filled better.

Validity of the findings

no comment

Additional comments

The paper is well-written.

Reviewer 3 ·

Basic reporting

All comments have been added in detail to the 4th section called additional comments.

Experimental design

All comments have been added in detail to the 4th section called additional comments.

Validity of the findings

All comments have been added in detail to the 4th section called additional comments.

Additional comments

Review Report for PeerJ Computer Science
(SLFCNet: an ultra-lightweight and efficient strawberry feature classification network)

Thanks for the revision. The revised paper and response letter were examined in detail. Considering the first report, which was stated in 9 items, it was observed that only a part of it was revised and an attempt was made to support it with various tables. All missing items in the first report, especially the item regarding evaluation metric, need to be answered and/or taken into consideration. In addition, the point-by-point response document must be prepared in detail and attached. I recommend that all items be taken into consideration and the relevant parts of the paper be revised again.

---

## Round 0.3 · accepted · Accept

Dear authors, we are pleased to verify that you meet the reviewer's valuable feedback to improve your research.

Thank you for considering PeerJ Computer Science and submitting your work.

Reviewer 1 ·

Basic reporting

All of the suggested reviews were revised.

Experimental design

All of the suggested reviews were revised.

Validity of the findings

All of the suggested reviews were revised.

Additional comments

All of the suggested reviews were revised.

Reviewer 3 ·

Basic reporting

All comments have been added in detail to the 4th section called additional comments.

Experimental design

All comments have been added in detail to the 4th section called additional comments.

Validity of the findings

All comments have been added in detail to the 4th section called additional comments.

Additional comments

Review Report for PeerJ Computer Science
(SLFCNet: an ultra-lightweight and efficient strawberry feature classification network)

Thanks for the revision. I have examined in detail the responses to the reviewer comments and the relevant changes. Some answers are limited but acceptable. Having reviewed this latest version and its contribution to the literature, I recommend that this paper be accepted. I wish the authors success in their future work. Best regards.